# Molecular Characterization and Pathogenesis of H6N6 Low Pathogenic Avian Influenza Viruses Isolated from Mallard Ducks (*Anas platyrhynchos*) in South Korea

**DOI:** 10.3390/v14051001

**Published:** 2022-05-08

**Authors:** Kaliannan Durairaj, Thuy-Tien Thi Trinh, Su-Yeon Yun, Seon-Ju Yeo, Haan-Woo Sung, Hyun Park

**Affiliations:** 1Zoonosis Research Center, Department of Infection Biology, School of Medicine, Wonkwang University, Iksan 570-749, Korea; kmdurairaj@gmail.com (K.D.); tndus2142@naver.com (S.-Y.Y.); 2Institute of Endemic Diseases, Medical Research Center, Department of Tropical Medicine and Parasitology, Seoul National University, Seoul 03080, Korea; trinhthithuytien.k56@hus.edu.vn; 3Department of Tropical Medicine and Parasitology, Department of Biomedical Sciences, College of Medicine, Seoul National University, Seoul 03080, Korea; 4College of Veterinary Medicine, Kangwon National University, Chuncheon 24341, Korea

**Keywords:** avian influenza virus, H6N6, South Korea, *Anas platyrhynchos*

## Abstract

The subtype H6N6 has been identified worldwide following the increasing frequency of avian influenza viruses (AIVs). These AIVs also have the ability to bind to human-like receptors, thereby increasing the risk of animal-human transmission. In September 2019, an H6N6 avian influenza virus—KNU2019-48 (A/Mallard (*Anas platyrhynchos*)/South Korea/KNU 2019-48/2019(H6N6))—was isolated from *Anas platyrhynchos* in South Korea. Phylogenetic analysis results revealed that the hemagglutinin (HA) gene of this strain belongs to the Korean lineage, whereas the neuraminidase (NA) and polymerase basic protein 1 (PB1) genes belong to the Chinese lineage. Outstanding internal proteins such as PB2, polymerase acidic protein, nucleoprotein, matrix protein, and non-structural protein belong to the Vietnamese lineage. Additionally, a monobasic amino acid (PRIETR↓GLF) at the HA cleavage site; non-deletion of the stalk region (residue 59–69) in the NA gene; and E627 in the PB2 gene indicate that the KNU2019-48 isolate is a typical low-pathogenic avian influenza (LPAI) virus. The nucleotide sequence similarity analysis of HA revealed that the highest homology (97.18%) of this isolate is to that of A/duck/Jiangxi/01.14 NCJD125-P/2015(H6N6), and the amino acid sequence of NA (97.38%) is closely related to that of A/duck/Fujian/10.11_FZHX1045-C/2016 (H6N6). An in vitro analysis of the KNU2019-48 virus shows a virus titer of not more than 2.8 Log10 TCID _50_/mL until 72 h post-infection, whereas in the lungs, the virus is detected at 3 dpi (days post-infection). The isolated KNU2019-48 (H6N6) strain is the first reported AIV in Korea, and the H6 subtype virus has co-circulated in China, Vietnam, and Korea for half a decade. Overall, our study demonstrates that Korean H6N6 strain PB1-S375N, PA-A404S, and S409N mutations are infectious in humans and might contribute to the enhanced pathogenicity of this strain. Therefore, we emphasize the importance of continuous and intensive surveillance of the H6N6 virus not only in Korea but also worldwide.

## 1. Introduction

Avian influenza viruses (AIVs) belong to the *Orthomyxoviridae* family and are composed of eight single-stranded negative-sense RNA segments. The AIVs have been identified based on the antigenic properties of the hemagglutinin (HA) and neuraminidase (NA) glycoproteins. AIVs are categorized into 18 HA (H1–16 in wild birds and H17–18 in bats) and 11 NA (N1–9 in wild birds and N10–11 in bats) subtypes [1,2]. Furthermore, AIVs have been classified into the following two groups based on their virulence: low-pathogenic AIVs (LPAIVs) and high-pathogenic AIVs (HPAIVs). Previous studies have revealed that AIVs can have an epidemic emergence in birds, pigs, horses, and humans [3,4,5].

Downie and Laver reported in 1973 that the H6 subtype AIVs were isolated in 1965 in Massachusetts in the United States [6]. Following this discovery, the H6 subtype of AIVs has been frequently identified in isolates obtained from poultry birds, wild birds, and domestic birds around the world [7]. Over the past two decades, surveillance studies indicate that H6 subtypes have become enzootic in wild birds of southern China, mostly as H6N6 isolates. These H6 subtypes were identified by the genomic transmission of their diversity from Eurasia and North American migratory birds [8]. Simultaneously, Cheon et al. (2018) reported that many AIVs (H1N8, H1N1, H4N6, H6N2, H6N1, and H6N8) and related non-structural genes originated from a North American lineage due to intercontinental exchanges during the 2012–2017 surveillance period in South Korea [9]. However, AIV strains H6N6 and H6N2 were found co-circulating in Southern China during the surveillance periods from 2009 to 2011, and both strains showed enhanced binding to α-2,6-linked sialic acids, linked with augmented viral replication in MDCK cells [10,11,12].

Globally, H6 subtype AIVs were divided into the following two major clades: (i) The first group clade mainly consisted of H6N2 and H6N6 AIVs from Asian regions; (ii) The second group clade was mostly composed of H6N1 and H6N2 AIVs from Asian regions and other regions around the world [13]. Huang et al. found that dominant H6N2 strains were replaced with H6N6 AIVs [14]. These two AIVs strains were coexist in live bird markets in several provinces of Southern China and contain human-like receptors that are transmitted by direct contact with wild birds and guinea pigs [15].

The aim of this study is to perform a phylogenetic and mutational analysis of H6N6 AIVs isolated from mallard ducks in South Korea in 2019. Moreover, to the best of our knowledge, there are no reports related to the pathogenetic analysis of South Korean H6N6 AIVs; thus, for the first time, we elucidate the complete pathogenesis of South Korean H6N6 AIVs through in vitro and in vivo experimental infections in a mammalian model. 

## 2. Materials and Methods

### 2.1. Sample Collection

During the surveillance period from January 2019 to December 2019, a total of approximately 4253 wild-bird fresh fecal samples were collected from the fields in Gyeonggi-do, South Korea (Latitude—37°23’46.8” N and Longitude—127°56’08.9” E) Figure 1. The collected samples were stored at 2–8 °C and transferred to the laboratory within 12 h for further analysis.

### 2.2. Virus Isolation from Feces

Fecal samples were processed according to our previously described protocol [16]. The collected fecal sample was dissolved in PBS (phosphate buffered saline; pH 7.4) supplemented with an antibiotic solution (100 U/mL penicillin and 100 mg/µL of streptomycin) (Merck, St. Louis, MO, USA) and centrifuged at 3000 rpm for 10 min at 4 °C. The supernatant was filtered through a hydrophilic polyethersulfone membrane filter having a pore size of 0.45 µm (GVS Syringe, Novatech, Kingwood, TX, USA) to remove bacteria and eukaryotic cells. The filtered supernatant samples were inoculated into the allantoic cavities of 10-day-old SPF (specific pathogen-free) ECEs (embryonated chicken eggs). Next, eggs were incubated at 37 °C for 96 h under humidified conditions, and their status was checked every day. Subsequently, eggs were chilled overnight at 4 °C, and allantoic fluids were harvested and analyzed for the confirmation of viruses based on HA (hemagglutination) activity using chicken erythrocytes. 

### 2.3. Bird Species Identification Using the Mitochondrial Gene Cytochrome c Oxidase I (COI) as a DNA Barcode

Host bird species were determined by confirmation of a DNA barcode of a region consisting of 751 base pairs (bp) of the mitochondrial gene cytochrome c oxidase I (COXI) as previously described elsewhere [17]. The identification of the host was discovered using information from Barcode of Life Data system (BOLD; Biodiversity Institute of Ontario, University of Guelph, ON, Canada) in a combination comparison with Basic Local Alignment Search Tool for nucleotides (BLASTn; NCBI, National Institute of Health, Bethesda, MD, USA).

### 2.4. Extraction of Viral RNA for Sequencing

A NucleoSpin RNA kit (MACHEREY-NAGEL, Düren, Germany) was used to extract the viral RNA directly from the allantoic fluid of ECEs according to the manufacturer’s instructions. Finally, the extracted RNA was eluted in RNase-free water distributed with 20 units of RNase inhibitor and was stored at −80 °C for further analysis. Conventional real-time RT-PCR was performed to determine the presence of the influenza virus and its subtypes using total RNA following WHO (World Health Organization) guidelines [17]. 

### 2.5. Next Generation Sequencing (NGS) Analysis

NGS analysis was accompanied by GnCBIO (Dae-Jeon, Korea) on the Illumina Hiseq X platform method as previously reported [16]. Briefly, the extracted viral RNA quality was determined using an Agilent RNA 6000 Pico kit (Agilent, Santa Clara, CA, USA), and total RNA concentration was measured using a nano spectrophotometer. The cDNA library of viral RNA was constructed using a QIAseq FX single-cell RNA library kit (Qiagen, Venlo, Netherlands). Library quality was evaluated via the LightCycler qPCR system (Roche, Penzberg, Upper Bavaria, Germany) and library size was verified using TapeStation HS D5000 ScreenTape system (Agilent, Santa Clara, CA, USA). For cluster generation, the library was loaded into a flow cell where fragments were seized on a lawn of surface-bound oligos complementary to the library adapters. Using bridge amplification, each fragment was amplified into distinct clonal clusters, following which cluster creation templates were ready to be sequenced. For sequencing, data were converted into raw data for analysis. Raw sequence reads were quality-trimmed using “trim galore” (q = 20), and non-influenza virus reads were excluded using Deconseq (iden = 60). The quantity of data was corrected using a Python script up to 600,000 reads. Ephemerally, a database of only segments 4-HA, 6-NA, and 8-NS1 from the influenza virus information from NCBI were generated and aligned to those of the reference using Gsmapper (iden = 70, mL = 40). The ORF (open reading frame) was observed using the obtained consensus and adopted a result with an ORF similar to the reference. As the ORF length varied from that of the reference, sequence error was altered using Proof Read as previously described [18].

### 2.6. Phylogenetic Analysis and Molecular Characterization

Nucleotide blast analysis was used to identify pertinency of the viral genes. Entire reference sequences were downloaded from the NCBI (National Center for Biotechnology Information https://www.ncbi.nlm.nih.gov/ (accessed on 27 March 2022)) and GISAID (Global Initiative on Sharing All Influenza Data https://www.gisaid.org/ (accessed on 27 March 2022)) databases. Data were merged, all duplicated sequences were deleted, and phylogenetic trees were finally constructed using MEGA6.0 software (Molecular Evolutionary Genetics analysis version 6.0, Pennsylvania State University, PA, USA). Phylogenetic trees of all segments (*PB2, PB1, PA, HA, NP, NA, M*, and *NS*) of the KNU2019-48 (H6N6) viral isolate were generated by applying the maximum-likelihood with Tamura-Nei model and 1000 bootstrap replicates.

### 2.7. Determination of 50% Tissue Culture Infectious Dose (TCID_50_) and 50% Egg Infectious Dose (EID_50_)

ELISA (enzyme-linked immunosorbent assay) was used to measure TCID_50_ titers as previously reported [19], MDCK cells (ATCC, Manassas, VA, USA) were grown on 96-well flat-bottom microplates at 37 °C with 5% CO_2_. MDCK cells with 80–90% confluence were washed with 1× concentrated PBS and then inoculated with serial 10-fold dilutions of virus suspensions in media containing 1 µg/mL of TPCK-trypsin. Virus-infected cells were incubated at 37 °C with 5% CO_2_ for 72 h. Next, the TCID_50_ titers were determined via the Reed and Muench method [20]. To determine EID_50_, the allantoic cavities of 10-day-old SPF ECEs were inoculated with 100 μL serial 10-fold dilutions of the viruses, using 5 eggs for each dilution. The eggs were incubated at 37 °C for 96 h. Allantoic fluid was harvested and tested using HA assays [21], and EID_50_ calculation of viruses was completed using the Reed and Muench method [20].

### 2.8. Viral Growth Kinetics in MDCK Cells

The growth kinetics of virus isolates were evaluated in vitro. MDCK cells were infected with the viruses at an MOI (multiplicity of infection) of 0.001 in DMEM medium containing 1% antibiotic and 1 μg/mL TPCK-trypsin. Supernatants were collected every 12 hpi up to 72 hpi. ELISA was performed using anti-influenza nucleoproteins (PAN antibodies) to detect infected cells, and the virus titer (TCID_50_) of each supernatant was determined [22].

### 2.9. Animal Experiment

The pathogenic potential of the new isolate was determined in mammals using 6-week-old female BALB/c mice purchased from Orient (Seongnam, Korea) (*n* = 11), which were intranasally infected with 10^5^ EID_50_/mL of virus. Mice were anesthetized using 1% isoflurane following the manufacturer’s instructions (Hana Pharmacy, Hwasung, Korea). The survival rate and bodyweight of mice were observed for 15 days, following which the mice were euthanized, and their lungs (*n* = 3 mice) were collected at 3, 6, and 15 dpi. The lung tissue was homogenized, and the TCID_50_ was determined to analyze the viral titers of homogenate supernatants [23]. For histopathology, the lungs of three mice were collected at 3, 6, and 15 dpi and stored in 4% formaldehyde/saline at 4 °C until further analysis. Hematoxylin and eosin (H&E) were used for histopathological examination of paraffin-embedded lung tissues mounted on glass slides. All sections were observed using a light microscope (magnification ×100). This study was approved by the Animal Ethics Committee of the Wonkwang University (WKU19-64; approved on 25 November 2019), South Korea, and all methods were conducted in accordance with relevant guidelines and regulations.

### 2.10. Statistics

The calculation of mean and SD (standard deviation) and Student’s *t*-test were performed using GraphPad Prism Software Version 5.0 (La Jolla, CA, USA). Results were presented as the mean ± SD. Statistical significance was set at *p* < 0.05.

## 3. Results

### 3.1. Genome Characterization of the KNU2019-48 (H6N6) Isolate

A/Mallard (*Anas platyrhynchos*)/South Korea/KNU 2019-48/2019(H6N6), designated as KNU2019-48 (H6N6), samples were isolated from feces of *Anas platyrhynchos* in South Korea on September 18, 2019. The genome sequence information of the isolate was deposited in GenBank (accession numbers MW380639 to MW380646). The GenBank accession numbers of the eight gene segments and the highest nucleotide identities from the GenBank database are shown in Table 1, with the sequence identities ranging from 96.09% to 99.21% when compared with our KNU2019-48 (H6N6) virus isolates. Surface gene HA was closely related to A/duck/Jiangxi/01.14 NCJD125-P/2015(H6N6), which originated from China, while NA was closely related to A/duck/Fujian/10.11_FZHX1045-C/2016 (H6N6) with nucleotide identities of 97.18% and 97.38%, respectively. PB2, PA, NP, M, and NS were closely related to A/duck/China/330D17/2018 (H6N6), which originated from China, with a nucleotide identity of 98.75, 99.44, 99.21, 97.59, and 98.86%, respectively. Similarly, the gene for PB1 was closely related to A/duck/Guangdong/7.20_DGCP015-C/2017 (H6N6), which originated from China, with a nucleotide similarity of 99.12%. Figure 2 shows the phylogenetic analysis of the KNU2019-48 (H6N6) strain for eight gene segments.

### 3.2. Hypothesis for the Reassortment Event of Each Gene Segment

Using evolutionary reassortment tracking analysis of our isolate, HA gene re-assortment prevailed from South Korea with links to China A/duck/Jiangxi/01.14 NCJD125-P/2015(H6N6) isolates in 2015. This was followed by the transmission of the HA gene from South Korea through the A/mallard/Korea/M219/2014 (H6N2) isolate in 2014. Similarly, based on the 2014–2015 report, the H6N6 strain NA gene co-circulated for a long time in China, which was later transmitted to South Korea. Next, the PB1 gene was transformed from Chinese isolates (A/duck/Guangdong/7.20_DGCP015-C/2017 (H6N6)).

The other genes (PB2, PA, NP, M, and NS) were collectively reassorted from the Chinese isolate A/duck/China/330D17/2018 (H6N6) in 2018. The backbone of these isolates originated from the Vietnamese isolates of A/Muscovy duck/Vietnam/LBM755/2014 (H5N6) in 2014. Additionally, there might have been a major H6N6 transmission from China during the migration season of 2018–2019. The locations of the putative origins of genomic components of the KNU2019-48 (H6N6) strain are shown in Figure 3.

Based on the evolutionary information, whole gene segments of our isolate KNU2019-48 (H6N6) originated from Chinese H6N6 isolates. It is possible that the Asian birds migrated through the East Asia–Australasian Flyway. Detailed information on the evolutionary re-assortment is presented in Figure 4.

### 3.3. Molecular Characterization of the KNU2019-48 (H6N6) Isolate

The HA cleavage site of KNU2019-48 (H6N6) contained the PRIETR↓GLF (↓denotes the cleavage site) sequence with a single basic amino acid in the HA cleavage site, which revealed a low pathogenic feature of the isolated virus. This HA receptor-binding site (RBS) suggests that H6 isolates would have a high preference toward α-2,6-linked sialic acid, which is abundant in the upper respiratory tract of avian species. We compared the HA protein of our isolate with those of the following four other isolated H6N6 strains that originated from birds and swine: (1) a low-pathogenic H6N6 isolate from domestic ducks in central China, labeled as A/duck/Hubei/10/2010 (H6N6); (2) A/swine/Guangdong/K6/2010 (H6N6) isolates, which originated from domestic ducks following whole gene reassortment; (3) isolates from poultry markets in Southern China that are a potential threat to mammals, labeled as A/duck/China/A729-2/2011(H6N6); (4) novel North American-origin avian influenza A (H6N5) virus isolated from bean goose in South Korea in 2018, which was published by our team in 2020, labeled as A/Bean goose/South Korea/KNU18-6/2018(H6N5).

A reasonable analysis of HA RBS at positions 138, 186, 190, 226, and 228 (H3 numbering) shows no mutation in amino acid residues (Table 2). The single basic residue at the HA cleavage site, with no prominent mutation at the HA RBS, showed that our KNU2019-48 (H6N6) isolates and reference strains, except for Hunan-2011, were low pathogenic H6 viruses. Moreover, the NAs of the H6N6 isolates had no amino acid deletion in the stalk (59–69) regions (Table 2). Mutations in other internal genes, also presented in Table 3, signify an enhanced viral replication efficiency along with virulence in mammals. Particularly, three human host marker mutations (PB1-S375N, PA-A404S, and S409N) were observed in KNU2019-48 (H6N6) strains, along with the mutation S375N in the PB1 gene displayed in our old H6N5 isolates from Korea, which had originated from North America. KNU2019-48 (H6N6) showed variance in certain molecular aspects, such as mutations at Q368R of PB2, K328N of PB1, S37A of PA, T223I of NA, and N30D of M1, each contributing to the potentially increased virulence of KNU2019-48 (H6N6) in mammals. 

### 3.4. Growth Kinetics of KNU2019-48 (H6N6) Isolate in Mammalian Cell Culture

Since the molecular characterization shows reassortment via mutation, our viral isolate could increase replication efficiency; therefore, in vitro examination of the viral replication of our isolate along with other isolates was performed. To evaluate the growth kinetics of our isolate, human-origin viruses, A/California/07/2009 (H1N1) and H7N7, which are infectious to mammals, were used as controls. H1N1 replicated more efficiently in MDCK cells as compared to KNU2019-48 (H6N6) and H7N7 (Figure 5). Raw data of the TCID_50_ assay are shown in Appendix A.

### 3.5. Pathogenicity in Mice

We further examined the pathogenic potential of mice (6-week-old female BALB/c) that were intranasally infected with 10^5^ EID_50_/50 µL. The H1N1 and H7N7 strains were used as the controls for comparisons. The bodyweight of the mice infected with the positive control H1N1 strain decreased from 3 dpi and was gradually regained after 10 dpi, with the lowest weight (76.15 ± 1.96%) observed at 8 dpi. Similarly, our KNU2019-48 (H6N6) isolates and the H7N7 control strain demonstrated a stable body weight trend for 15 days (Figure 6A).

All the infected mice survived the entire experimental period and showed no difference in the survival mortality between the three groups (Figure 6B). Virulence in mice lungs after 3, 6, and 15 dpi were analyzed by TCID_50_ assay. Based on the experimental results, control group H1N1 showed maximum virus titer in the lungs in comparison with the KNU2019-48 (H6N6) and H7N7 strains at 3 dpi (4.62 ± 0.17, 3.76 ± 0.38, 2.71 ± 0.08 log_10_ TCID_50_/mL, respectively), and these viral loads gradually decreased at 6 dpi (4.26 ± 0.44, 3.80 ± 0.63, 1.80 ± 0.05 log_10_ TCID_50_/mL, respectively) (Figure 6C). In contrast, no virus was detected in all groups at 15 dpi, indicating that these viruses were not yet well-adapted for murine infections. Further, histopathological analyses were conspicuously shown in the lungs of the virus-challenged mice. Infected lungs at 3 and 6 dpi show a condensed penetration of neutrophils into the alveolar air spaces (Figure 7).

## 4. Discussion

The H6N6 subtype AIV is widespread in poultry, and its host range extends to mammals such as pigs. Moreover, it has become an endemic disease in domestic animals [61]. Our isolated KNU2019-48 (H6N6) strain was co-circulated in China, Vietnam, and Korea for half a decade. These H6N6 subtype gene segments were derived from the East Asia–Australasian flyway lineages. Poultry birds may play an intermediate host role in the cross-species transmission of the influenza virus from domestic birds to humans [62]. However, this is the first isolation of H6N6 strains during our 2018 surveillance program in South Korea, and there are no detailed evidential molecular and pathological studies on H6N6 isolates from South Korea. In particular, other subtypes such as H7N3 and other H7 subtypes of AIV were pre-vailed/isolated during the surveillance period [16,63].

According to the molecular analysis of the H6N6 strain in our study, the HA cleavage site sequence PRIETR↓GLF was subtyped into a low-pathogenic virus. According to the results of phylogenetic analysis, the HA gene was closely related to A/mallard/Korea/M219/2014 (H6N2), while the genes of PB2, PA, NP, M, and NS were closer to the Chinese and Vietnamese isolates (A/duck/China/330D17/2018 (H6N6) and A/Muscovy duck/ Vietnam/LBM755/2014 (H5N6)). The PB1 gene was similar to that of the Chinese isolate, A/duck/Guangdong/7.20_DGCP015-C/2017 (H6N6), and the NA gene was similar to that of co-circulated Chinese isolates A/duck/Fujian/10.11_FZHX1045-C/2016 (H6N6) and A/chicken/Guangdong/8.30_DGCP022-O/2017(H6N6), respectively. These analyses strongly indicate that the H6 subtype virus has been frequently detected in migratory ducks in China [48,64,65].

Generally, H6 isolates receptor-binding sites were composed of HA protein residues (Q226 and G228), which preferentially bind to the α-2,3-linked sialic acid receptors in avian bird species [61]. Similarly, based on previous reports, the following HA protein residues: A137N, P186L, A193N, and G228S, were associated with human receptor-binding preference and were not found in our H6N6 isolate [7,15]. Furthermore, H6N6 isolates can be modified to replace V187D in HA responses for the binding affinity might be adapted to mammalian receptors [66]. Mutation residues E119V, H275Y, R293K, and N295S were found in the NA gene, which suggested that H6 isolates could provide insights regarding oseltamivir inhibitors. No amino acid deletions or mutations were found in the NA of KNU2019-48 (H6N6). The KNU2019-48 (H6N6) isolate carries mutations in the proteins of PB2 (Q368R, H447Q) [29,30], PB1 (D/A3V, L13P, R207K, K328N) [29,30,35], PA (F277S, C278Q, S/A515T) [43,45], NA (T223I) [53,54], M1 (N30D, V15I/T, T215A) [53,54,55], and NS1 (A/P42S) [67], which led to adaptation and increased virulence in mammals. However, the well-known mammalian adaptation markers in the protein of PB2 (E627K, D701N) were not found in our isolate [68]. Nevertheless, the KNU2019-48 (H6N6) isolate revealed human-host markers in the PB1 (S375N/T) and PA (A404S, S409N) genes [29,30,44].

Furthermore, we re-confirmed the pathogenicity of our KNU2019-48 (H6N6) isolate using in vitro experiments. We monitored the growth kinetics of KNU2019-48 (H6N6) replication in MDCK cells as follows: the virus replicated promptly and exhibited no obvious signs of illness. A maximum virus titer (2.84 log_10_ TCID_50_/mL) was achieved at 24 hpi, and there were no changes until 72 h post-infection. Similarly, our isolate strongly exhibited low pathogenicity in the lungs of mice at 3 and 6 dpi compared to the positive control H1N1, which originated from humans. Additionally, we performed body weight and survival percentage analysis. Low pathogenicity was confirmed in intranasally-infected mice; mice lungs weighed the highest at 3 dpi, whereas at 6 and 15 dpi, no variance was observed, and the lung weight was normal (Figure 6D). Although our H6N6 isolate is a low-pathogenic isolate, its movement should be monitored frequently to detect incidences of infection in humans and a potential genetic reassortment event.

In conclusion, this is the first study that broadens the knowledge of Korean H6N6 isolates from evolution to mammalian cell expressions and in vivo characterization. Continuous monitoring and molecular characterization of the H6N6 virus will be required for a better understanding of the evolutionary dynamics of the virus, which can further assist in improving control measures.

## Figures and Tables

**Figure 1 viruses-14-01001-f001:**
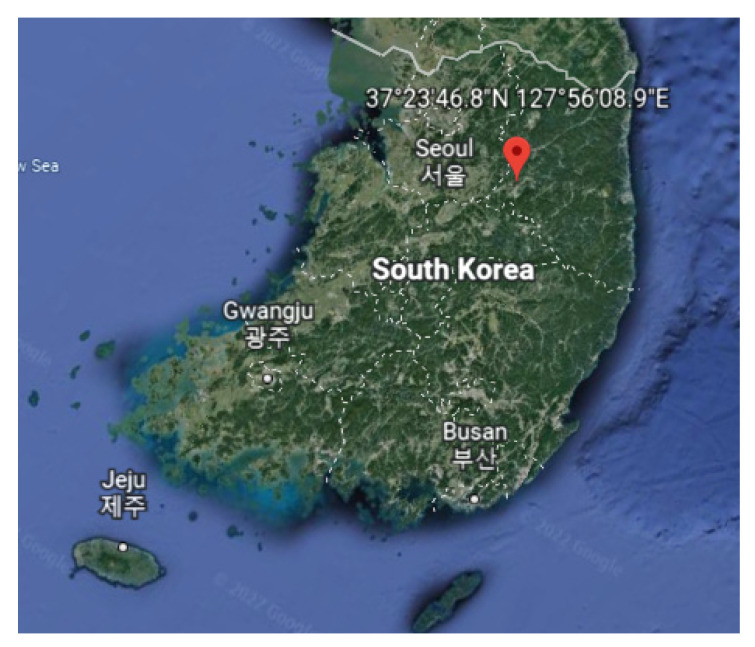
Google Earth image for sample colleting location with the information of latitude and longitude.

**Figure 2 viruses-14-01001-f002:**
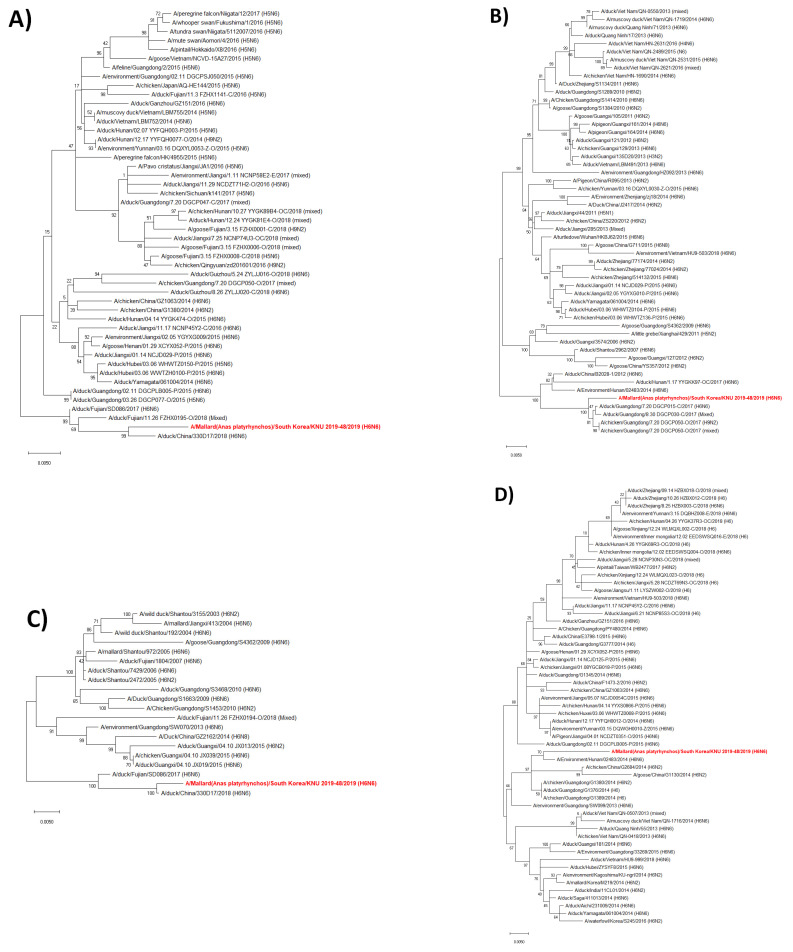
(**A**–**H**) Phylogenetic analysis of the KNU2019-48 (H6N6) strain for eight gene segments. (**A**) PB2; (**B**) PB1; (**C**) PA; (**D**) HA; (**E**) NP; (**F**) NA; (**G**) M; (**H**) NS. (PB—polymerase basic protein; NP—nucleoprotein; HA—hemagglutinin; PA—polymerase acidic protein; NA—neuraminidase; M—matrix protein; NS—non-structural protein).

**Figure 3 viruses-14-01001-f003:**
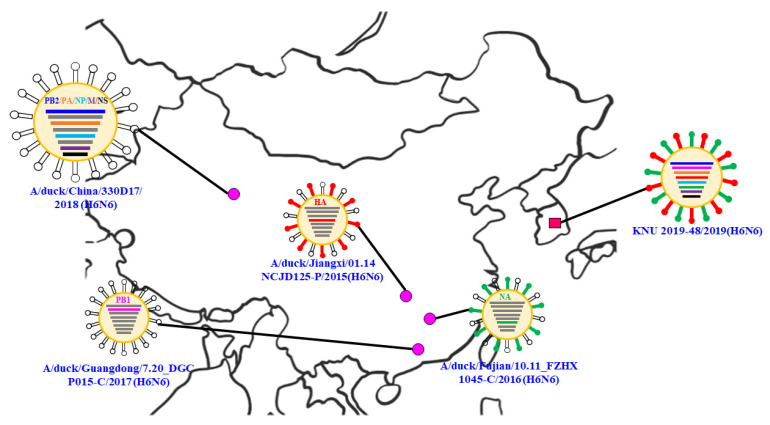
Locations of the putative origins of genomic components of the KNU2019-48 (H6N6) strain.

**Figure 4 viruses-14-01001-f004:**
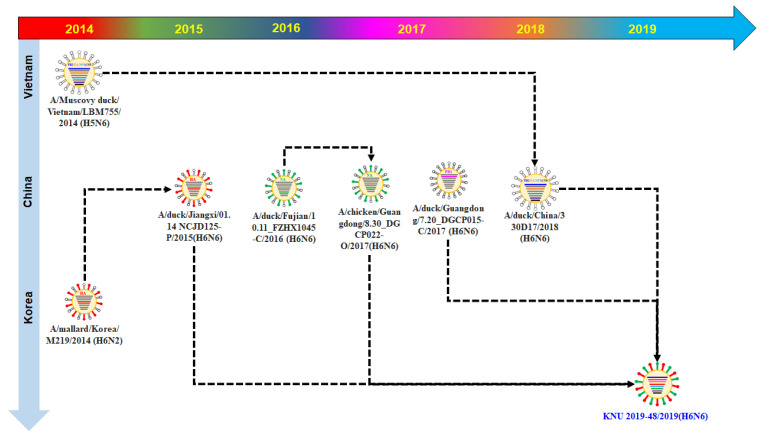
Original reassortment events of the novel avian influenza isolate KNU2019-48 (H6N6).

**Figure 5 viruses-14-01001-f005:**
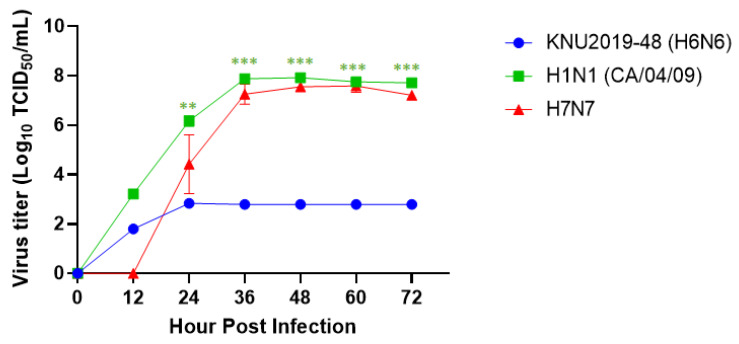
Growth kinetics of KNU2019-48 (H6N6) replication in MDCK cells. Three kinds of viruses were infected into MDCK cells at a multiplicity of infection (MOI) of 0.001. The cell supernatants were collected at different time-points (12, 24, 36, 48, 60, 72, and 84 hpi). The virus titer concentration in cell culture supernatant was determined by an enzyme-linked immunosorbent assay (ELISA) using anti-influenza nucleoprotein (NP) to detect infected cells, and TCID_50_ was determined in MDCK cells. Data are represented as mean ± SD and calculated from three replicates. **, *p* < 0.01; ***, *p* < 0.001.

**Figure 6 viruses-14-01001-f006:**
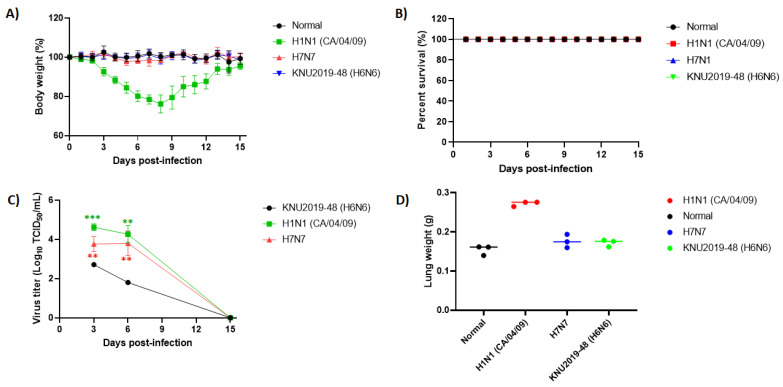
In vivo pathogenicity of the KNU2019-48 (H6N6) virus isolate. BALB/c mice were intranasally infected with 10^5^ EID_50_ concentrations of the virus per mouse. H1N1 and H7N7 virus isolates were used as control. Mean changes in (**A**) body weight, (**B**) survival rates, (**C**) virus titers in the lung, and (**D**) lung weight were noted. Body weights are presented as percentages of the original weight (*n* = 5). **, *p* < 0.01; ***, *p* < 0.001.

**Figure 7 viruses-14-01001-f007:**
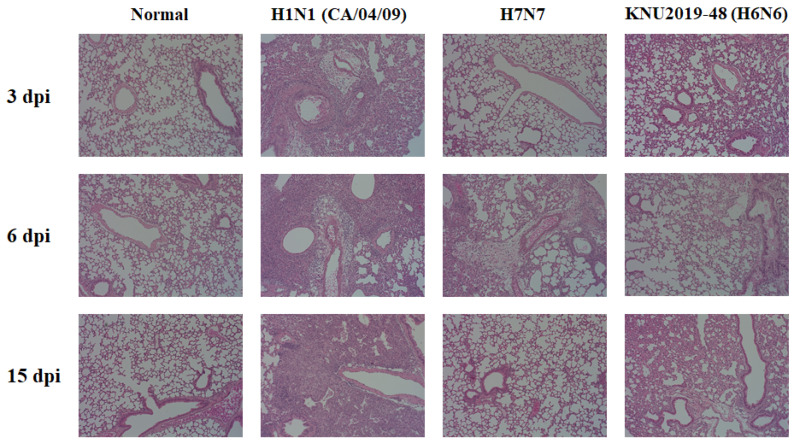
Histology of lung inflammation determined by hematoxylin and eosin (H&E) staining. For each isolate, BALB/c mice were intranasally infected with 10^5^ EID_50_ concentrations of the virus per mouse. The uninfected control (normal); KNU2019-48 (H6N6)-, H1N1 (CA/04/09)-, and H7N7-infected mouse lungs were collected and stained with H&E at days 3, 6, and 15 post infection (dpi) (scale bar, 100 µm; original magnification ×100).

**Table 1 viruses-14-01001-t001:** Virus strains obtained from the GenBank database with highest nucleotide identities when compared with the KNU2019-48 (H6N6) isolate in this study.

Gene	GenBank ID	Reference Strain Accession ID	Origin	Per Ident (%)
PB2	MW380639	EPI_ISL_501514	A/duck/China/330D17/2018 (H6N6)	98.75 (2341/2328)
		EPI_ISL_285466	A/duck/Fujian/SD086/2017 (H6N6)	98.29 (2280/2328)
		EPI666098	A/duck/Guangdong/02.11_DGCPLB005-P/2015 (H6N6)	97.16 (2335/2328)
PB1	MW380640	EPI_ISL_707456	A/duck/Guangdong/7.20_DGCP015-C/2017 (H6N6)	99.12 (2274/2304)
		MW104102	A/chicken/Guangdong/7.20_DGCP050-O/2017(mixed)	99.03 (2274/2304)
		EPI_ISL_698000	A/chicken/Guangdong/7.20_DGCP050-O/2017 (H9N2)	99.03 (2274/2304)
PA	MW380641	EPI_ISL_501514	A/duck/China/330D17/2018 (H6N6)	99.44 (2233/2151)
		EPI_ISL_285466	A/duck/Fujian/SD086/2017 (H6N6)	98.14 (2151/2151)
		EPI_ISL_76327	A/duck/Shantou/2472/2005 (H6N2)	96.09 (2151/2151)
HA	MW380642	EPI_ISL_199312	A/duck/Jiangxi/01.14 NCJD125-P/2015(H6N6)	97.18 (1740/1701)
		MH130170	A/mallard/Korea/M219/2014 (H6N2)	96.47 (1726/1701)
		EPI_ISL_219853	A/Environment/Hunan/02483/2014 (H6N6)	98 (1701/1701)
NP	MW380643	EPI_ISL_501514	A/duck/China/330D17/2018 (H6N6)	99.21 (1565/1527)
		EPI_ISL_696839	A/duck/Guizhou/10.28_ZYLJJ001-C/2018 (H6N6)	98.33 (1497/1527)
		MW098939	A/duck/Guangdong/7.20_DGCP030-C/2017(mixed)	97.33 (1497/1527)
NA	MW380644	EPI_ISL_696964	A/duck/Fujian/10.11_FZHX1045-C/2016 (H6N6)	97.38 (1412/1465)
		EPI666988	A/duck/Guangxi/04.10_JX019/2015 (H6N6)	96.10 (1412/1465)
		MW100376	A/chicken/Inner_mongolia/12.02_EEDSWSQ002-C/2018 (H6N6)	95.47 (1413/1465)
M	MW380645	MN088783	A/duck/China/330D17/2018 (H6N6)	97.59 (1027/979)
		MW101275	A/duck/Fujian/11.26_FZHX0181-C/2018(mixed)	99.18 (976/979)
		LC028304	A/muscovy duck/Vietnam/LBM755/2014(H5N6)	99.18 (976/979)
NS	MW380646	MN088790	A/duck/China/330D17/2018(H6N6)	98.86 (890/889)
		MW101859	A/duck/Guizhou/10.28_ZYLJJ001-C/2018(H6N6)	97.75 (844/889)
		CY109470	A/duck/Shantou/17490/2006(H6N2)	97.16 (844/889)

**Table 2 viruses-14-01001-t002:** Comparison between the HA receptor-binding sites and NA of the avian influenza H6N6 virus and other host-pathogenic H6 viruses.

Virus Strains	HA Receptor-Binding Residues (H3 Numbering)						NA				
	Cleavage Sites 340-348	A138S	P186L	E190V	Q226L	G228S	Stalk Region Deletion	E119V	H275Y	R293K	N295S
KNU2019-48 (H6N6)	PRIETR↓GLF	A	P	E	Q	G	NO	E	H	R	N
H10/2010 (H6N6)	PQIETR↓GLF	A	P	E	Q	G	NO	E	H	R	N
K6/2010 (H6N6)	PQIETR↓GLF	S	I	E	Q	S	NO	E	H	R	N
A729-2/2011 (H6N6)	PQIETR↓GLF	A	P	E	Q	G	YES (59–69)	E	H	R	N
KNU18-6/2018 (H6N5)	PQIETR↓GLF	A	P	E	Q	G	NO	E	H	R	N

KNU2019-48 (H6N6): A/Mallard (*Anas platyrhynchos*)/South Korea/KNU 2019-48/2019(H6N6), H10/2010 (H6N6): A/duck/Hubei/10/2010 (H6N6), K6/2010 (H6N6): A/swine/Guangdong/K6/2010 (H6N6), A729-2/2011 (H6N6): A/duck/China/A729-2/2011(H6N6), KNU18-6/2018 (H6N5): A/Bean goose/South Korea/KNU18-6/2018(H6N5). ↓ cleavage site.

**Table 3 viruses-14-01001-t003:** Summary of data obtained from the mutational analysis of eight genes from AIVs of multiple avian species with the KNU2019-48 (H6N6) isolate.

Viral Protein	Amino Acid	KNU2019-48 (H6N6)	H10/2010 (H6N6)	K6/2010 (H6N6)	A72-2/2011 (H6N6)	KNU18-6/2018 (H6N5)	Phenotype	Reference
PB2	K147T, M147L	I	I	I	I	I	-	[24]
	T63I (with PB1 M677T)	I	I	I	I	I	Pathogenic in mice	[25]
	L89V	V	V	V	V	V	Enhanced polymerase activity; Increased virulence in mice	[26]
	K251R	R	R	R	R	R	Increased virulence in mice	[27]
	I292V	I	I	I	I	I	Increase the polymerase activity of H7N9 viruses in both avian and human cells and facilitate the transmission	[28]
	G309D	D	D	D	D	D	Enhanced polymerase activity; Increased virulence in mice	[26]
	T339K	K	K	K	K	K	Enhanced polymerase activity; Increased virulence in mice	
	Q368R	R	R	R	R	R	Increased polymerase activity; Increased virulence in mammals	[29,30]
	H447Q	Q	Q	Q	Q	Q	Increased polymerase activity; Increased virulence in mammals	
	I471T (with PB2 P453H)	T	T	T	T	T	Change the surface electrostatic potential drastically	[31]
	R477G	G	G	G	G	G	Enhanced polymerase activity; Increased virulence in mice	[26]
	I495V	V	V	V	V	V	Enhanced polymerase activity; Increased virulence in mice	
	A676T	T	T	T	T	T	Enhanced polymerase active; Increased virulence in mice	
	E627K	E	E	E	E	E	Mammalian adaptation marker	[32,33]
	D701N	D	D	D	D	D	Mammalian adaptation marker	
PB1	D/A3V	V	V	V	V	V	Increased polymerase activity; Increased virulence in mammals	[29,30]
	L13P	P	P	-	P	P	Increased polymerase activity; Increased virulence in mammals, Mammalian host marker	[34]
	R207K	K	K	K	K	K	Increased polymerase activity in mammalian cells	[35]
	K328N	N	N	N	N	N	Increased polymerase activity; Increased virulence in mammals	[29,30]
	S375N/T	N	N	N	N	N	Increased polymerase activity; Increased virulence in mammals, Human host marker	
	H436Y	Y	Y	Y	Y	Y	Increased polymerase activity and virulence in mallards, ferrets, and mice	[36]
	A469T (with NS1 N205K; NEP T48N)	T	T	T	T	T	Conferred in contact transmissibility in guinea pigs	[35]
	L473V	V	V	V	V	V	Increased polymerase activity and replication efficiency	
	V652A	A	A	A	A	A	Increased virulence in mice	[27]
	M677T (with PB2 T63I)	I	I	I	V	I	Pathogenic in mice	[25]
	V598P	L	L	L	L	L	Decreased polymerase activity and replication efficiency in mammalian cells	[37,38]
	D622G	G	G	G	G	G	Increased polymerase activity and virulence in mice	[39]
PA	N383D	D	D	D	D	D	Increased polymerase activity in mammalian and avian cell lines	[40,41]
	S37A	A	A	A	A	A	Significantly increased viral growth and polymerase activity in mammalian cells	[42]
	H266R	R	R	R	R	R	Increased polymerase activity; Increased virulence in mammals and birds	[43]
	F277S	S	S	S	S	S	Adapt to mammalian hosts	
	C278Q	Q	Q	Q	Q	Q	Adapt to mammalian hosts	
	I357K	T	T	T	T	T	Increased polymerase activity; Increased virulence in mammals and birds	
	N383D (with S224P)	D	D	D	D	D	Enhanced the pathogenicity and viral replication of H5N1 virus in mice	[40,41]
	A404S	S	S	S	S	A	Human host marker	[44]
	S409N	N	N	N	N	S	Enhanced Transmission; Human host marker	
	S/A515T	T	T	T	T	T	Increased polymerase activity; Increased virulence in mammals and birds	[43,45]
	L653P	P	P	P	P	P	Adapt to mammalian hosts	[43]
HA	V110A	A	A	A	A	A	Host specificity shift to Enhance binding of HA to human-type SAα2,6Gal receptor	[46]
	T160A	E	G	S	A	E	Increased binding to human-type influenza receptor	[47]
	T/E173G/D/V	D	D	D	D	T	Increased virus binding to α-2,6-linked sialic acid	[22,48]
NP	V41I	I	I	I	I	I	Might contribute to viral transmissibility	[49]
	V105M	M	I	M	M	M	Contribute to the increased virulence of the H9N2	[50]
	D210E	E	E	E	E	E	Might contribute to viral transmissibility	[49]
	F253I	I	I	I	I	I	Results in attenuated pathogenicity of the virus in mice	[42]
	A286V	A	A	A	A	A	Affect the pathogenicity of the virus in mice	[51]
	I353V	V	V	V	V	V	Increased virulence in mice	[27]
	T437M	T	T	T	T	T	Affect the pathogenicity of the virus in mice	[51]
NA	M26I	V	I	I	I		Increased virulence in mice	[52]
	T223I	I	I	I	I		Increased virulence in mammals	[53,54]
M1	N30D	D	D	D	D	D	Increased virulence in mammals	[55]
	V15I/T	I	I	I	I	I	Increased virulence in mammals	[53,54]
	A166V	V	V	V	V	V	Contribute to the increased virulence of the H9N2.	[49]
	T215A	A	A	A	A	A	Increased virulence in mammals	[55]
NS1	A/P42S	S	S	S	S	S	Increased virulence in mammals; Antagonism of IFN induction	[56]
	T80E	N	T	T	T	T	Reduced influenza virus replication through controlling RIG-I-mediated IFN production and vRNP activity	[57]
	T/D/V/R/A127N	D	N	N	N	N	Increased virulence in mammals	[58]
	V149A	A	A	A	A	A	Pathogenicity in mice; Antagonism of IFN induction	[59]
NS2	T47A (with NS1 N200S)	E	E	E	E	E	Decreased IFN antagonism	[60]
	M51I (with NS1 G205R)	R	R	R	R	R	Decreased IFN antagonism	

(“-“. no phenotypes were found).

## Data Availability

All the data presented in this study are available in this article and Appendix A.

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
