# Peer review of "Molecular Characterization and Pathogenesis of H6N6 Low Pathogenic Avian Influenza Viruses Isolated from Mallard Ducks (Anas platyrhynchos) in South Korea"

_viruses, 2022, doi:10.3390/v14051001_

Round 1

Reviewer 1 Report

Surveillance and study of new variants of highly pathogenic avian influenza viruses is a very important and urgent task. We all know the high danger of H5 and H7 variants of highly pathogenic influenza virus. The presented article showed the potential danger of the H6N6 subtype of avian influenza virus for both domestic birds and humans. The work was done at a high methodological level, well illustrated and carries new information about this variant of avian influenza. This article will be interesting for virologists, veterinary specialists and epidemiologists. I believe that it can be published without changes.

Author Response

Reviewer – 1

Surveillance and study of new variants of highly pathogenic avian influenza viruses is a very important and urgent task. We all know the high danger of H5 and H7 variants of highly pathogenic influenza virus. The presented article showed the potential danger of the H6N6 subtype of avian influenza virus for both domestic birds and humans. The work was done at a high methodological level, well-illustrated and carries new information about this variant of avian influenza. This article will be interesting for virologists, veterinary specialists and epidemiologists. I believe that it can be published without changes.

Response) All the authors are grateful to the anonymous reviewers for indicating merits and demerits of our study and their valuable comments for accepting appropriate journal.

Reviewer 2 Report

Review report is attached 

Author Response

Reviewer – 2

I reviewed the manuscript ID viruses-1678586 -entitled “Molecular characterization and pathogenesis of H6N6 low pathogenic avian influenza viruses isolated from mallard ducks (Anas platyrhynchos) in South Korea” for the journal of MDPI viruses

Overall Comments to authors:

Overall, this manuscript lacks detail, ethical statement, and correct molecular analysis, which detracts from the overall paper. I suggest revising the introduction. The current form of introduction lacks detail, coherence, and the significance of their research, gap analysis appropriate references, which detracts from this research's overall objectives.

Response) All the authors are grateful to the anonymous reviewers for indicating merits and demerits of our study and their valuable comments and suggestions. As per the reviewers’ comments, we have revised in the whole manuscript and the corrections were highlighted in the revised manuscript.

Specific comments:

The authors did not mention any information related to ethics statements and the facility to conduct their research. they should write about ethical approval to conduct the animal experimental study.

Response) We appreciate your suggestion, we had included the information accordingly and highlighted in the revised version.

Revise)

Page 5 Line 173-176: “This study was approved by the Animal Ethics Committee of the Wonkwang University (WKU19-64; December 19, 2019), South Korea, and all methods were conducted in accordance with relevant guidelines asnd regulations.”

Page 17 Line 381-383: “Institutional Review Board Statement: This study was conducted according to the guidelines of the Animal Ethics Committee of Wonkwang University (WKU19-64, approval on 25 November 2019).”

Line 74-76: how many samples did they collect? if these are environmental fecal samples, then how did they confirm the fecal samples were from Malad ducks. Did they perform DNA barcoding? Duration of study? I suggest adding sampling locations and characteristics to the map

Response) We added the information of the surveillance location map, duration, and sample amount, accordingly.

Revise)

Page 2-3 Line 75-81: “During the surveillance period from January 2019 to December 2019, a total of ap-proximately 4253 wild-bird fresh fecal samples were collected from the fields in Gyeong-gi-do, South Korea (Latitude - 37°23'46.8"N and Longitude - 127°56'08.9"E) Figure 1.”

Figure 1. Google earth image for sample colleting location with the information of latitude and longitude

Response) We also added the methodology for bird identification by PCR of DNA barcoding gene COI.

Revise)

Page 3 Line 95-103:

“2.3. Bird species identification using the mitochondrial gene cytochrome c oxidase I (COI) as a DNA barcode

Host -bird species was determined by confirmation of a DNA barcode of a region consisted of 751 base-pairs (bp) of the mitochondrial gene cytochrome c oxidase I (COXI) as previously described elsewhere.

The identification of the host was discovered using information from Barcode of Life Data system (BOLD; Biodiversity Institute of Ontario, University of Guelph, ON, Canada) in a combination comparison with Basic Local Alignment Search Tool for nucleotides (BLASTn; NCBI, National Institute of Health, Bethesda, MD, USA).”

The prevalence of AIV and other subtypes including H6N6

Response) As per the reviewer suggestions, we have included the prevalence information in the revised manuscript.

Revise)

Page 18 Line 319-320:

In particular, other subtypes like H7N3, and other H7 subtypes of AIV were prevailed/isolated during the surveillance period.

Line 78-79: replace the stool sample with the fecal sample.

Response) Thank you for your correction. We had re-write the word accordingly.

Revise)

Page 2 Line 83-86: “Fecal samples were processed according to our previously described protocol. The collected feces fecal sample was dissolved in PBS (Phosphate buffered saline; pH 7.4) supplemented with an antibiotic solution (100 U/mL penicillin and 100 mg/µL of streptomycin) (Merck, St. Louis, MO, USA) and centrifuged at 3000 rpm for 10 min at 4 °C.”

Line 122-127: I suggest changing the phylogenetic analysis of the neighbor-joining method Regenerate the phylogenetic trees using a maximum-likelihood method to clarify the evolution of the H6N6-subtype AIVs

Response) As per your suggestion, we have regenerated the phylogenetic trees of the H6N6-subtype AIVs using a maximum-likelihood method and we have inserted in to the revised version.

Revise)

Page 4 Line 139-141: Phylogenetic trees of all segments (PB2, PB1, PA, HA, NP, NA, M, and NS) of the KNU2019-48 (H6N6) viral isolate were generated by applying the maximum-likelihood with Tamura-Nei model and 1000 bootstrap replicates.

Page 9 Line 200-203: Figure 2. (A–H) Phylogenetic analysis of the KNU2019-48 (H6N6) strain for eight gene segments. (A) PB2; (B) PB1; (C) PA; (D) HA; (E) NP; (F) NA; (G) M; (H) NS. (PB - polymerase basic protein; NP - nucleoprotein; HA - hemagglutinin; PA - polymerase acidic protein; NA - neuraminidase; M - matrix protein; NS - non-structural protein).

Generate HA tree with lineage information. I suggest performing time to the most recent common ancestor analyses.

Response) As per your suggestion, we have tried to generate HA tree with lineage information through BEAST v1.10.4. For this initially, we have calculated the time signal and Root-to-plot regression of the estimated sequences through the IQ-TREE server (http://iqtree.cibiv.univie.ac.at/) and TempEst v1.5.3, respectively. The result of Root-to-plot of genetic distance against the time of HA sequences regression (R2=0.1933) is very low and the major drawback there was not much report about H6 strains after 2015 and also here we have attached the TempEst v1.5.3 result. Due to these reasons, we cannot able to present HA tree lineage information.

Results, and Conclusions, as in their current state, these sections seem to be scattered throughout the manuscript. The authors should recommend to the future direction and the implications of their research.

Response) As per the reviewer suggestion, we have mentioned the future direction into the result and conclusion part and highlighted in revised version.

Revise)

Page 18 Line 362-365: Continuous monitoring and molecular characterization of the H6N6 virus will be re-quired for a better understanding of the evolutionary dynamics of the virus, which can further assist in improving control measures.
